# The Application of Box–Behnken Design for Investigating the Supercritical CO_2_ Foaming Process: A Case Study of Thermoplastic Polyurethane 85A

**DOI:** 10.3390/molecules29020363

**Published:** 2024-01-11

**Authors:** Salal Hasan Khudaida, Shih-Kuo Yen, Chie-Shaan Su

**Affiliations:** Department of Chemical Engineering and Biotechnology, National Taipei University of Technology, Taipei 10608, Taiwan; salalhassan@mail.ntut.edu.tw (S.H.K.);

**Keywords:** supercritical CO_2_ foaming, thermoplastic polyurethane, Box–Behnken design

## Abstract

Thermoplastic polyurethane (TPU) is a versatile polymer with unique characteristics such as flexibility, rigidity, elasticity, and adjustable properties by controlling its soft and hard segments. To properly design and understand the TPU foaming process through supercritical CO_2_, a design of experiments approach, the Box–Behnken design (BBD) was adopted using commercial TPU 85A as the model compound. The effect of saturation pressure, saturation temperature, and immersion time on the mean pore size and expansion ratio were investigated. The design space for the production of TPU foam was shown, and the significance of process parameters was confirmed using the analysis of variance (ANOVA). In addition, extrapolation foaming experiments were designed and validated the feasibility of the response surface model developed via BBD. It was found that the pore size of TPU 85A foam could be controlled within 13 to 60 μm, and a stable expansion ratio could be designed up to six.

## 1. Introduction

Polymeric foam is a material that contains a large number of gas pores in its structure and shows specific characteristics such as lightweight, low density, excellent thermal insulation, high specific strength, high impact strength, and a low dielectric constant [1]. It can be applied in various fields, including automotive industry, packaging industry, aerospace, building construction, sporting goods, and biomedical applications. Since the applications of polymeric foams are versatile, developing an efficient and environment-friendly foaming process is crucial to control foam properties such as the pore size, cell density, and expansion ratio. To minimize the concerns related to adopting chemical foaming agents, such as the requirement of high-temperature conditions and contamination in the produced foams, physical foaming agents are more favorable, especially for producing the foam material used as household goods or biomedical materials. In recent decades, using supercritical fluid, especially for supercritical CO_2_, as the foaming agent, has been an alternate way to replace chemical foaming agents used in conventional processes. The supercritical foaming process provides advantages, including no release of toxic gases, no use of organic solvents, and a better control of foam properties [2,3,4]. For example, Haurat and Dumon fabricated porous polymers from macro to nano porosities in batch or continuous supercritical CO_2_ foaming processes [5]. By adopting a two-step depressurization batch process, Bao et al. successfully used supercritical CO_2_ as a foaming agent to design polystyrene (PS) foams with dual pore structures. Their results can help estimate the bi-modal cell morphology of PS and other polymers synthesized via a two-step depressurization batch process [6]. Yang et al. investigated the supercritical CO_2_ foaming of radiation cross-linked isotactic polypropylene in the presence of triallylisocyanurate. Their study demonstrated that radiation cross-linking improves foamability and promotes the formation of precisely defined cell structures [7]. Dai et al. designed thermoplastic polyurethane microcellular composite fibers by extrusion using supercritical CO_2_ as a foaming agent. They used nanoclay particles as the nucleating agent in a supercritical CO_2_-based single filament extrusion foaming technique to create TPU fibers with micron to submicron cell size. The use of modified montmorillonite clay nanoparticles (CloisiteTR 20A) in the matrix at an appropriate concentration of about 1.0 wt% increased the nucleation rate of foaming, resulting in the development of tiny bubbles in the extruded fibers [8]. Jahani et al. studied the high expansion effect of open-cell polycarbonate foams and investigated the effect on sound insulation, thermal conductivity, and mechanical properties, then developed lightweight polycarbonate foams that can be used for sound insulation and thermal insulation [9].

Thermoplastic polyurethane (TPU) is a multiblock copolymer that contains hard and soft segments. By modifying the content or type of the hard and soft segments, the properties of TPU, such as rigidity, hardness, flexibility, and elastomeric behavior, can be efficiently manipulated. In addition, TPU is a thermo-processable material. Its waste was recyclable and available to process into a secondary product and was treated as an environmentally friendly polymer. Microcellular TPU foams are very interesting materials for various industries, such as the furniture, automotive, sportswear, packaging, electronics, and medical devices industries, since they show excellent performance, such as outstanding abrasion resistance, high strength, high toughness, and well-defined elasticity. Due to these excellent properties and potential applications, several studies have been reported in the literature to design TPU microcellular foams. For example, Wang et al. developed structure-tunable TPU foams and investigated their mechanical properties [10]. Nofar et al. studied the foaming mechanism of TPU containing different hard and soft segments. In their studies, they demonstrated that by increasing the hard segment’s content and raising the soft segment’s molecular weight, the shrinkage effect in TPU foams was significantly reduced [11,12]. Jiang et al. prepared sheets of TPU foam using a mold foaming process and investigated the foaming mechanism at different foaming temperatures. The soft TPU foam sheets obtained in their work displayed outstanding resilience and good dimensional stability, durability, and shock absorption capability [13]. Yeh et al. prepared the nanocellular TPU foams using nanoclay as the nucleation agent and investigated the mechanism for producing nanocellular TPU foams. They demonstrated that by utilizing Cloisite 30B nanoclay (Clay 30B) and graphene nanoparticles as nucleation agents, they produced nanocellular foam. Additionally, they provided new insights into solid-state foaming [14,15]. Prasad et al. investigated the effect of polymer hardness, pore size, and porosity on TPU-based chemical mechanical polishing (CMP) pad performance [16]. They demonstrated that the solid-state microcellular foaming (SSMF) process can manufacture CMP pads with various pore sizes and porosities, employing a broad range of TPU hardness resins. Zhang et al. used the supercritical CO_2_ technique to create expanded thermoplastic polyurethane (ETPU) beads, which were then compressed to form foam sheets. They demonstrated that the capabilities of compression molding ETPU to create foam sheets have been confirmed. However, the processing window for obtaining foam sheets without deforming the cell structure in the ETPU during the sintering of ETPU beads is relatively narrow [17].

To efficiently design and manipulate the properties of TPU foam to meet the application’s requirements, a systematical investigation of the effect of foaming parameters is essential, especially by adopting the design of experiments (DOE) approach. Even though the foaming of TPU by supercritical CO_2_ has been demonstrated in the literature, the DOE study for a comprehensive understanding of the foaming parameter was still scarce, especially for hard TPU [18,19]. Therefore, this study adopted a design of experiments approach, the Box–Behnken design (BBD), to investigate the supercritical CO_2_ foaming process for a commercial and hard TPU with a Shore hardness of 85A. The equations represent the response surface of the mean pore size, and the expansion ratio was proposed. In addition, the significance of the operating parameters, including the high-order effect, was verified and discussed using the analysis of variance (ANOVA). Furthermore, according to the developed response surface, the validating foaming experiments were designed, and the feasibility of controlling the TPU foam properties was finally demonstrated.

## 2. Results

In this study, a BBD was adopted to systematically investigate the effect of operating parameters and construct the response surface for the supercritical CO_2_ foaming of TPU. The BBD is cost-effective and efficient for a wide range of variables compared to other response surface approaches, such as the three-level factorial design. Before the BBD investigation, a screening of the appropriate intervals of operating parameters was required, and several foaming experiments (Exps. A1 to A12) were conducted, as listed in Table 1. The expansion ratio and pore structure of the produced TPU foam are also shown in Table 1 and Figure 1. As can be seen, increasing the saturation pressure from 80 to 120 bar (Exps. A1 to A3) has a moderate effect on producing TPU foam with a high expansion ratio. In addition, several experiments were conducted with different immersion times (2–4 h) to study their effect on foaming. SEM analysis showed that increasing immersion time did not significantly affect overall pore structure. Thus, the effect of immersion time was negligible according to the results obtained from Exps. A1 and A9, Exps. A5 and A10, Exps. A6 and A11, and Exps. A3 and A12. On the other hand, the effect of saturation temperature was considerable and exhibited an optimized condition (Exp. A1, A4 to A8). The results indicated that at temperatures from 60–80 °C, the material had difficulty foaming, and pore growth was poor. As the temperature increased to 100–140 °C, the pore size gradually increased due to the growing state. The maximum pore size is reached at 140 °C. However, operating at the highest saturation temperature of 160 °C led to the collapse of the pore structure due to the material melting, resulting in the disappearance of pores.

According to the screening foaming experiments results in Table 1 and Figure 1, the independent variables and levels employed in this BBD study were summarized in Table 2. The saturation pressure and saturation temperature were designed within 80–120 bar and 100–140 °C, respectively. Although the effect of immersion time seems insignificant according to the results listed in Table 1, a longer immersion time was designed as 6 h in the BBD study for confirmation. Table 3 lists the conditions and outcomes for 17 foaming experiments in BBD investigation with five replicates at the center point. Two responses, the expansion ratio and the mean pore size of the TPU foam, were selected. Three different models, including linear, two-factor interaction (2FI), and quadratic, were analyzed individually to screen the model for representing the responses. The models were considered based on the *R*^2^ and *p*-values of the lack-of-fit test to determine the significance of each model [20,21]. Model adequacy is presented in Table 4. When the results of the models were compared, it was obvious that the quadratic model had the highest *R*^2^ for data fitting, indicating that the quadratic model can adequately represent the variations in response. Furthermore, the *p*-values of the lack-of-fit test for the models can provide additional insights into selecting the best model [22]. In the lack-of-fit test, the *p*-values for the quadratic model regarding the expansion ratio and mean pore size were 0.0814 and 0.1702, respectively. Since both *p*-values are higher than 0.05, the lack of fit when applying the quadratic model is considered insignificant. Consequently, the quadratic model was chosen to fit the present data.

The experimental expansion ratio and mean pore size were correlated by adopting the quadratic model, and the empirical equations were listed in Table 5. The variables A, B, and C represent the saturation pressure in bar, saturation temperature in °C, and immersion time in hr. The average absolute relative deviations (AARD) for the experimental and calculated responses are also given in Table 5. It can be concluded that the quadratic model developed in the BBD study is available to represent the response surface of the expansion ratio and mean pore size of TPU foam. The AARDs of the expansion ratio and mean pore size are 7.1% and 6.0%, respectively.

Since the quadratic model can successfully represent the response surface of the expansion ratio and mean pore size, the operating conditions for designing TPU foam with specific properties through supercritical CO_2_ foaming can be evaluated. To validate the extrapolation ability, two validation foaming experiments, listed in Table 6, were designed, and the operating conditions were evaluated using the equations reported in Table 5 for producing TPU foam with a high expansion ratio and small (Exp. C1) or large (Exp. C2) mean pore size. According to Table 6, the experimental and predicted properties, including the expansion ratio and mean pore size, show good agreement. The quadratic models developed in this BBD study are feasible for predicting the properties of TPU foam extrapolative. The SEM images for the designed TPU foam obtained from Exp. C1 and Exp. C2 are presented in Figure 2. According to Table 6 and Figure 2, with a stable expansion ratio of about six, the mean pore size of TPU foam can be manipulated from 13 to 60 μm.

## 3. Discussion

In addition to developing the empirical equations to represent the relationship between responses and independent variables, the significance of the model term was also evaluated using an ANOVA to discuss the effect of the operating parameter. Table 7 lists the *p*-values of each model term in the quadratic model. If the *p*-value for a model term is less than 0.05, this term could be considered significant statistically to the response. According to Table 7, the effect of immersion time was negligible within 2 to 6 h. The immersion of TPU in a short time period, such as 2 h, is sufficient to reach a CO_2_ saturation condition. To confirm the negligible effect of immersion time from the BBD experiments, there are four sets of foaming experiments designed to compare the immersion time from 2 to 6 h, including Exps. B9 and B4 at the immersion conditions of 120 bar and 120 °C, Exps. B6 and B10 at the immersion conditions of 100 bar and 100 °C, Exps. B11 and B15 at the immersion conditions of 80 bar and 120 °C, and Exps. B7 and B13 at the immersion conditions of 100 bar and 140 °C. From Table 3, the expansion ratio, pore size, and cell density among each set were similar. In addition, Figure 3 compares the SEM images of the TPU foam obtained from the above experimental sets. As can be seen, the effect of immersion time was confirmed to be negligible.

According to Table 7, the most significant model term for the expansion ratio is the effect of saturation temperature (B) and its second-order term (B^2^). Both the *p*-values of model terms B and B^2^ are less than 0.0001. It indicates that the effect of saturation temperature is significant and shows a quadratic behavior, as shown in Figure 4. At the investigated interval, the expansion ratio increases with the saturation temperature, and the increase trend becomes considerable at the high-temperature region. Nofar et al. and Wang et al. reported a similar trend for the effect of saturation temperature on the expansion ratio of TPU [10,11,12]. In general, increasing saturation temperature facilitates cell growth and leads to an increased expansion ratio of TPU foam. In addition, the melt strength of the TPU foam decreases as the saturation temperature increases. The decrease in melt strength at higher saturation temperatures benefits the foaming process. This is because reduced melt strength may promote better cell growth, significantly improving the expansion ratio. The expanded foam probably has more gaps or cells, making it lighter and potentially more suited for applications requiring lightweight materials with insulating qualities. Furthermore, decreased melt strength indicates the molten TPU is less resistant to deformation and expansion. Although other model terms are insignificant according to the result of ANOVA, these model terms are still required to add to the empirical equation to support the model hierarchy.

Regarding the mean pore size, the effect of saturation pressure (A) and its second-order term (A^2^) were significant, as shown in Figure 5. Both the *p*-values of model terms B and B^2^ are less than 0.0001. Obviously, the mean pore size decreased with the increase in saturation pressure, and this effect became negligible in the high-pressure region. According to the experimental evidence reported by Li et al. and Primel et al. [23,24], the solubility of CO_2_ in TPU is proportional to the saturation pressure. Therefore, the increased CO_2_ solubilization with saturation pressure contributes to the increase in nucleation sites, resulting in a reduction in the mean pore size of the TPU foam. However, the numerous generated nucleation sites may collide and merge, and the pressure effect on the mean pore size under high-pressure conditions may be eliminated. In addition to the saturation pressure, the contribution of the effect of saturation temperature (B) on the mean pore size was also significant (*p*-value < 0.0001). The mean pore size increased as the saturation temperature increased, as presented in Figure 6. According to the literature solubility information [23,24], the solubility of CO_2_ dissolved in TPU would decrease as the saturation temperature increased, resulting in fewer nucleation sites in foaming. In addition, a high saturation temperature also facilitates cell growth during expansion. These mechanisms may result in a mean pore size increase with the saturation temperature.

In addition to the mentioned main effects, according to Table 7, a cross-interaction effect of saturation temperature and saturation pressure (AB) on the mean pore size was also confirmed to be significant (*p*-value = 0.0023). As presented in Figure 7, operating at a condition that combined the low saturation pressure and high saturation temperature shows a considerable promotion effect for generating the large pore size in the TPU sample. In this condition, a low CO_2_ solubilization, a low melt strength, and a favorable condition for cell growth may contribute to the production of TPU foam with a large pore size. Finally, it is worth mentioning that our study employs ANOVA and relies on *p*-values (Table 7) to assess the significance of model terms, in line with Wasserstein and Lazar’s ASA statement on *p*-values [25]. Transparency in reporting addresses concerns of misuse. Considering ASA’s emphasis on nuanced interpretations, especially for saturation temperature, future studies should incorporate effect sizes and confidence intervals. Addressing multiple comparisons, as recommended by ASA, would enhance robustness. In summary, our study aligns with ASA’s transparency recommendations and guides future research for the improved interpretability of results.

## 4. Materials and Methods

The TPU raw material, polyester-based TPU pellets with a Shore hardness value of 85A, were purchased from Golding Precision Material Co. (EB-85AR1). CO_2_, used as the foaming agent, was purchased from Yu Sheng Gas Co., Ltd. with a minimum purity of 99.5%. According to our previous study, a heat treatment before the foaming process was beneficial for producing TPU foam with a lower shrinkage ratio [26]. Therefore, the received TPU pellets were dried at 80 °C for 16 h then processed via compounding using an extruder at 165 °C. Approximately 50 mg of TPU were cut into a cube with a length of 0.4 cm from the compounded sample and dried at 80 °C for 16 h to remove moisture. The dried TPU sample was used to perform the supercritical CO_2_ foaming experiments. In addition, the thermal properties of polymeric material are crucial for selecting the foaming method and immersion temperature. Differential scanning calorimetry (DSC) and thermogravimetric analysis (TGA) are standard approaches for identifying the thermal properties of polymers [27,28,29,30,31]. The decomposition temperatures of TPU materials were first identified using TGA (PerkinElmer, Pyris 1). TPU samples weighing 5 to 10 mg were heated from 50 °C to 650 °C at a rate of 20 °C/min under a N_2_ gas environment. Figure 8 shows that the weight loss via the decomposition of TPU starts at around 350 °C and stabilizes at around 470 °C. The glass transition temperature (*T*_g_) and melting point (*T*_m_) of TPU were characterized using DSC (NETZSCH, 200 F3). Approximately 5 to 10 mg of TPU were measured in a N_2_ environment. The process involved three temperature changes in total. The thermal history of the material was initially eliminated using the heating-cooling-heating method, and then the material’s thermal properties were measured. The thermal conditions were programmed in three stages: the first stage of heating ranged from room temperature to 220 °C, the second stage of cooling ranged from 220 °C to −80 °C, and the third stage of heating ranged from −80 °C to 220 °C. As illustrated in Figure 9, TPU exhibited a glass transition *T*_g_ at around −22 °C and a melting temperature of around 152 °C. Since a low *T*_g_ was confirmed by DSC, a one-step foaming method was adopted in this study. Furthermore, the maximum saturation temperature was determined at 160 °C according to the melting temperature. Regarding the selection of immersion time, in the literature, Gunasekaran et al. measured the CO_2_ sorption behavior for a TPU from 20 to 40 min [32]. Michaeli and Heinz showed that the sorption of CO_2_ in a TPU reaches a maximum absorption amount at about 50 min [33]. Belmonte et al. investigated the supercritical CO_2_ foaming for four TPU samples with the immersion time ranging from 1 to 3 h [34]. Accordingly, this study chose a 2-h immersion time as the central point condition.

The experimental apparatus for the supercritical CO_2_ foaming experiment is shown in Figure 10. The experimental setup mainly consists of a CO_2_ supply section (1), a CO_2_ pressurization section (2), a temperature control section (3), and a chiller (4). The TPU samples were placed inside the high-pressure vessels (V-1 to V-3) and immersed in an oil bath to maintain the saturation temperature. The volume of each high-pressure vessel for the foaming experiment is 5 cm^3^. A high-pressure syringe pump was used to pressurize CO_2_ to a desired pressure. The TPU sample was immersed in the supercritical CO_2_ environment for a given immersion time. After the immersion step for CO_2_ dissolving into TPU, the pressure in each high-pressure vessel was immediately released through the three-way ball valves (B-1 to B-3), and the depressurization time was about 1 s. The produced TPU foam was then immersed into a chiller at 0 °C to stabilize the cell structure. For a given foaming condition, three foaming experiments were performed in parallel in three individual high-pressure vessels (V-1 to V-3) to confirm the reproducibility of the supercritical CO_2_ foaming process.

Regarding the physical property analysis of produced TPU foam, the expansion ratio of TPU foam was calculated by determining the $\rho_{p}$ and $\rho_{f}$, which are the densities of the TPU sample before and after foaming. The $\rho_{p}$ and $\rho_{f}$ were determined using a water-displacement method according to ASTM D792. After foaming, TPU samples may exhibit shrinkage or dimensional changes. Therefore, we allowed the TPU samples to rest for 5 days to stabilize and reduce the possibility of further shrinkage. Considering the shrinking behavior of the TPU foam, $\rho_{f}$ was determined with time, and the stable expansion ratio was finally reported in this study. The expansion ratio of TPU foam is calculated using the following equation.
(1)$$Expansion\;ratio=\frac{\rho_{p}}{\rho_{f}}$$

In addition, a scanning electronic microscope (SEM) was used to observe the cellular morphology of the foamed sample. The TPU sample was cryo-fractured in liquid nitrogen and then sputter-coated with gold for SEM analysis. The pore size of TPU foam was calculated using Image J software(JSM-IT100). Furthermore, with the known values of $\rho_{p}$ and $\rho_{f}$, the number of cells in the SEM images, and the actual measuring area of the SEM picture, the cell density of the TPU foam can be determined through the following equation.
(2)$$D={\left(\frac{N}{A}\right)}^{1.5}\times (\frac{\rho_{p}}{\rho_{f}})$$
where *D*, *N*, and *A* are the cell density in cells/cm^3^, the number of cells in the SEM images, and the actual measuring area of the SEM picture in cm^2^, respectively.

## 5. Conclusions

A design of experiments approach, the BBD, was adopted in this study to investigate the supercritical CO_2_ foaming process using a commercial TPU with a Shore hardness value of 85A as the model compound. The relationship between the expansion ratio and mean pore size was described appropriately with the quadratic model. The extrapolation for the developed models for designing TPU foam with the specific properties was validated. According to the ANOVA results, the significant effects of the supercritical CO_2_ foaming process were reported and discussed. The saturation temperature can efficiently control the expansion ratio of TPU foam. Combining the adjustment of saturation temperature and pressure, the mean pore size of the TPU foam can be successfully manipulated.

## Figures and Tables

**Figure 1 molecules-29-00363-f001:**
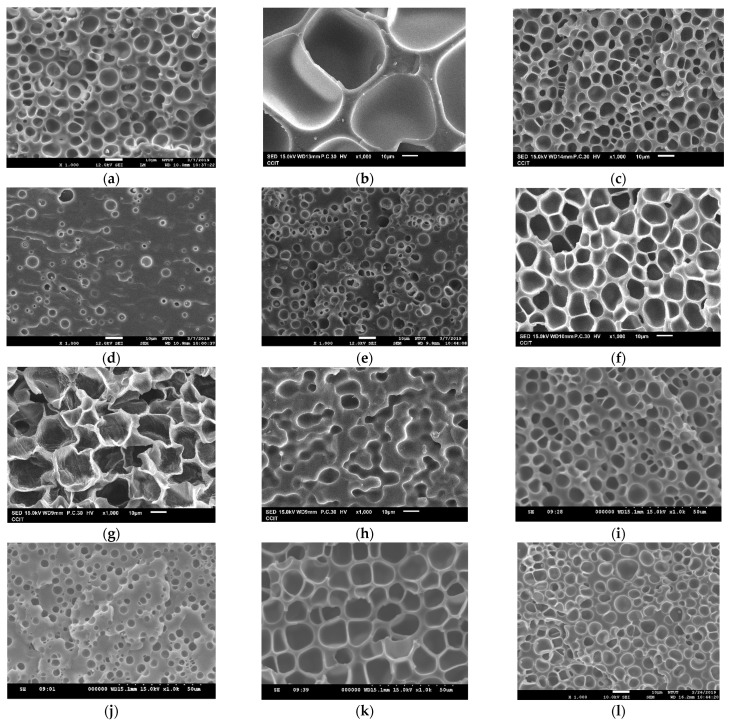
Comparison of the pore morphology of TPU foam obtained from (**a**) Exp. A1, (**b**) Exp. A2, (**c**) Exp. A3, (**d**) Exp. A4, (**e**) Exp. A5, (**f**) Exp. A6, (**g**) Exp. A7, (**h**) Exp. A8, (**i**) Exp. A9, (**j**) Exp. A10, (**k**) Exp. A11, and (**l**) Exp. A12.

**Figure 2 molecules-29-00363-f002:**
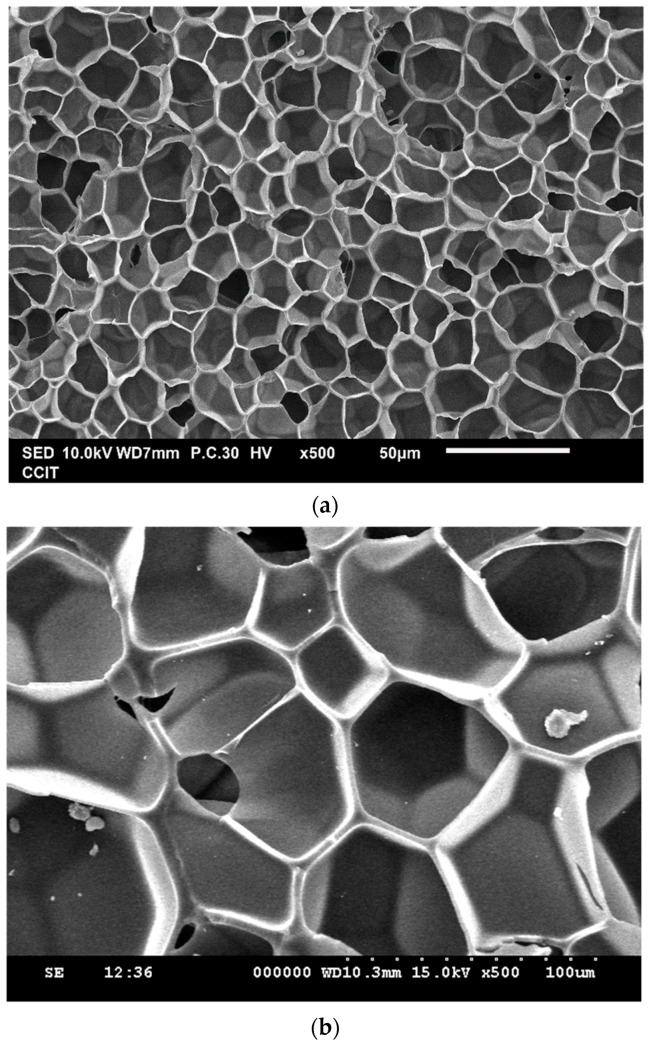
SEM images for TPU foam obtained from (**a**) Exp. C1 and (**b**) Exp. C2.

**Figure 3 molecules-29-00363-f003:**
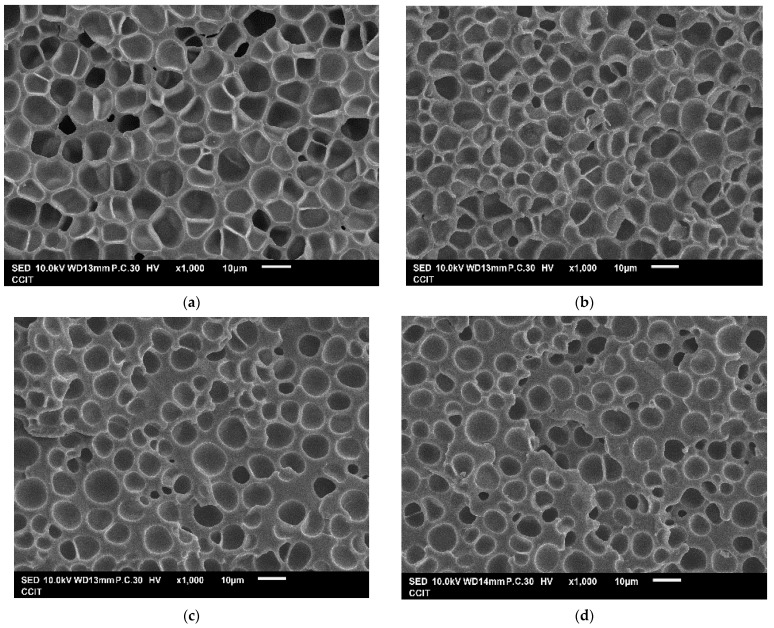
SEM images for TPU foam obtained from (**a**) Exp. B9, (**b**) Exp. B4, (**c**) Exp. B6, (**d**) Exp. B10, (**e**) Exp. B11, (**f**) Exp. B15, (**g**) Exp. B7, and (**h**) Exp. B13.

**Figure 4 molecules-29-00363-f004:**
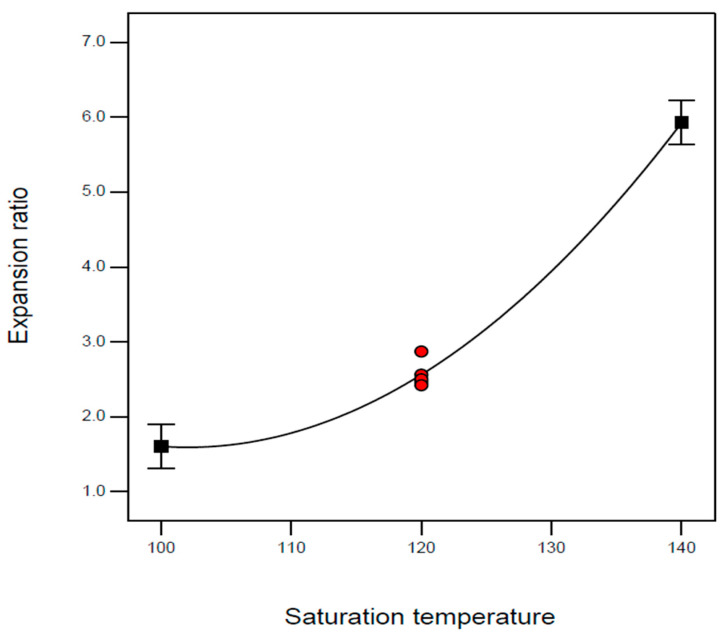
Effect of saturation temperature on the expansion ratio of TPU foam at saturation pressure of 100 bar and immersion time of 4 h.

**Figure 5 molecules-29-00363-f005:**
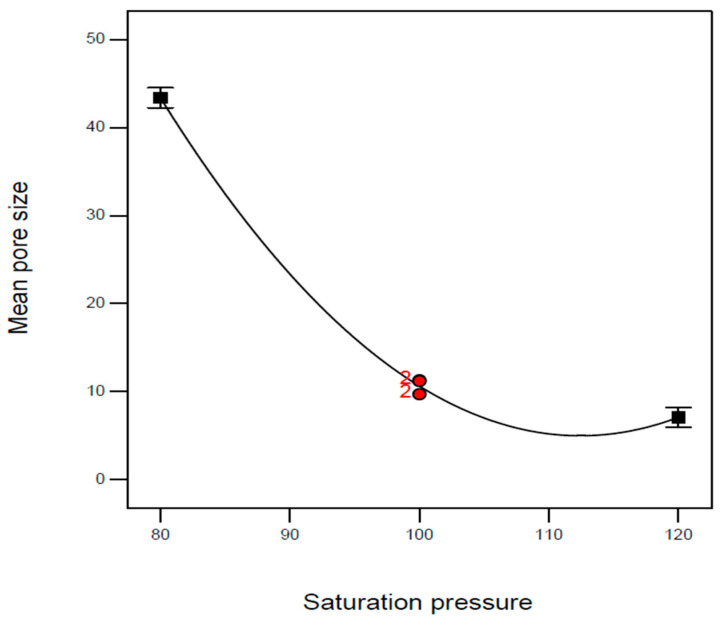
Effect of saturation pressure on the mean pore size of TPU foam at a saturation temperature of 120 °C and immersion time of 4 h.

**Figure 6 molecules-29-00363-f006:**
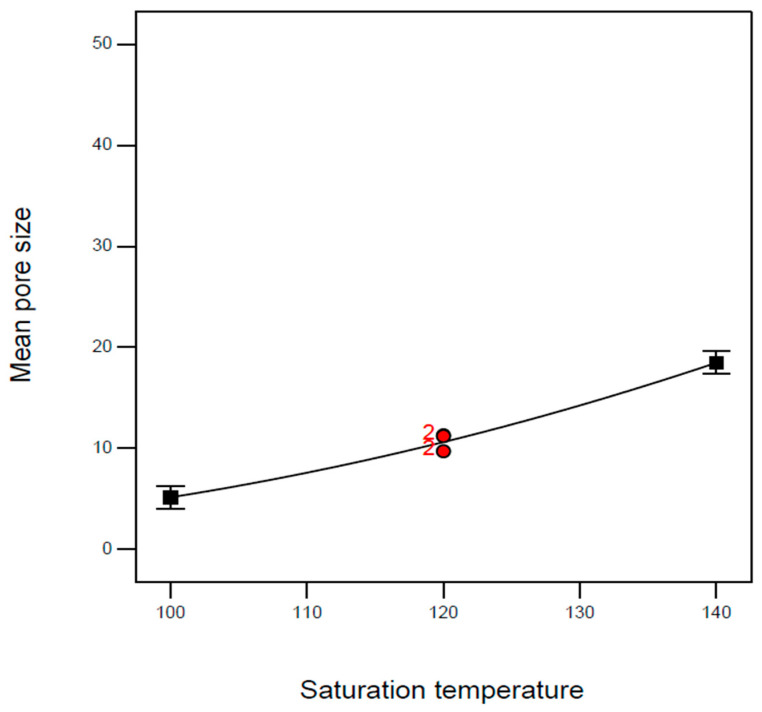
Effect of saturation temperature on the mean pore size of TPU foam at saturation pressure of 100 bar and immersion time of 4 h.

**Figure 7 molecules-29-00363-f007:**
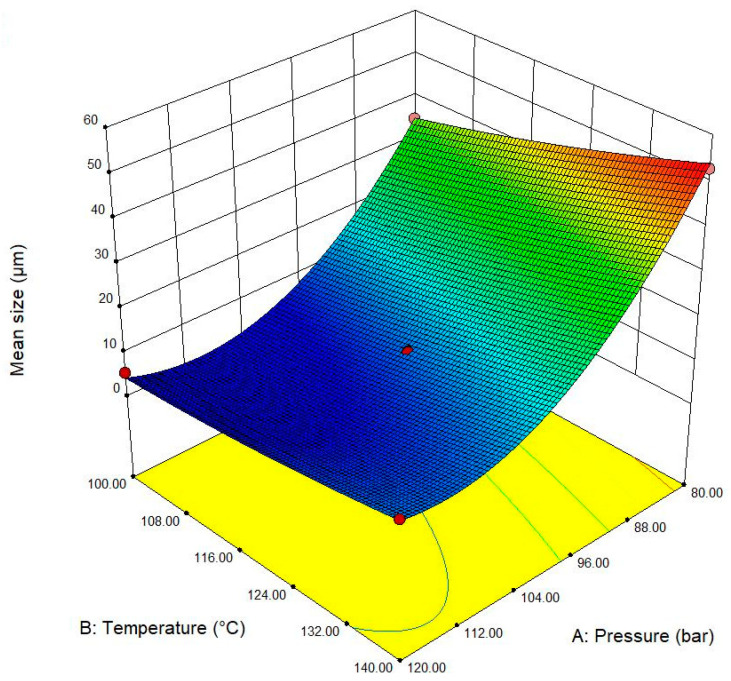
Graphical presentation of the interaction effect of saturation pressure and saturation temperature (AB) on the mean pore size of TPU foam.

**Figure 8 molecules-29-00363-f008:**
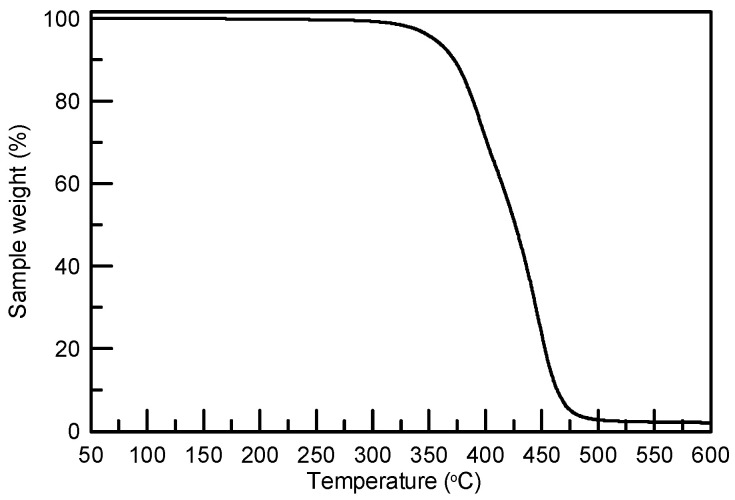
TGA results for unprocessed TPU 85A.

**Figure 9 molecules-29-00363-f009:**
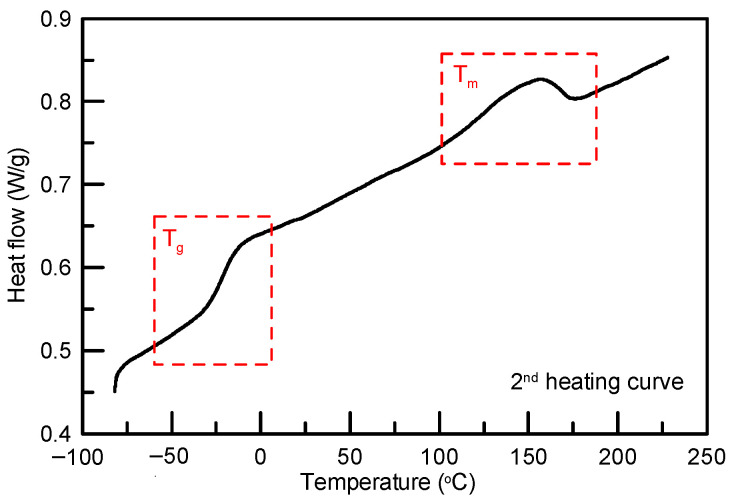
DSC thermograms for unprocessed TPU 85A.

**Figure 10 molecules-29-00363-f010:**
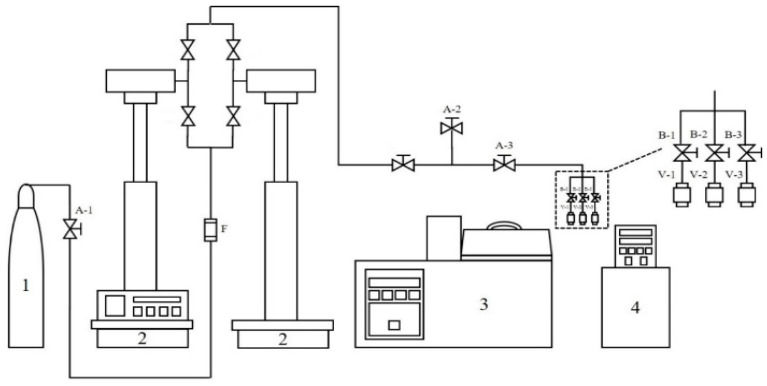
Experimental apparatus of supercritical CO_2_ foaming (1: CO_2_ cylinder, 2: syringe pump system, 3: thermostatic oil bath, 4: circulating chiller, V: high-pressure vessel, F: filter, A: two-way needle valve, B: three-way ball valve).

**Table 1 molecules-29-00363-t001:** Experimental conditions and results for screening the operating intervals in the BBD investigation.

Exp. No.	*P* (bar)	*T* (°C)	*t* (h)	Expansion Ratio (-)
A1	100	100	2	1.81
A2	80	100	2	1.38
A3	120	100	2	2.02
A4	100	60	2	1.08
A5	100	80	2	1.30
A6	100	120	2	3.18
A7	100	140	2	5.44
A8	100	160	2	1.29
A9	100	100	4	1.83
A10	100	80	4	1.26
A11	100	120	4	3.07
A12	120	100	4	1.85

**Table 2 molecules-29-00363-t002:** The control factors and levels for the BBD study.

Independent Variable	Symbol	Level		
		−1	0	1
A: Saturation pressure	*P* (bar)	80	100	120
B: Saturation temperature	*T* (°C)	100	120	140
C: Immersion time	*t* (hr)	2	4	6

**Table 3 molecules-29-00363-t003:** Experimental condition and results for investigating the supercritical CO_2_ foaming of TPU 85A by BBD investigation.

Exp. No.	Operating Parameter			Results		
	*P* (bar)	*T* (°C)	*t* (hr)	Exp. Ratio (-)	Pore Size (μm)	Cell Density (cells/cm^3^)
B1	100	120	4	2.87	11.3	1.67 × 10^9^
B2	100	120	4	2.42	11.2	1.40 × 10^9^
B3	80	140	4	6.00	52.7	2.38 × 10^7^
B4	120	120	6	2.92	7.8	3.97 × 10^9^
B5	100	120	4	2.56	9.7	1.34 × 10^9^
B6	100	100	2	1.56	6.7	1.76 × 10^9^
B7	100	140	2	5.88	22.6	5.95 × 10^8^
B8	100	120	4	2.50	9.7	1.67 × 10^9^
B9	120	120	2	2.86	9.2	3.87 × 10^9^
B10	100	100	6	1.53	7.2	1.87 × 10^9^
B11	80	120	2	2.03	47.0	1.10 × 10^7^
B12	120	140	4	5.43	12.7	3.51 × 10^9^
B13	100	140	6	6.22	19.5	9.88 × 10^8^
B14	80	100	4	1.35	34.9	7.18 × 10^6^
B15	80	120	6	2.09	45.7	1.22 × 10^7^
B16	120	100	4	1.80	5.4	6.68 × 10^9^
B17	100	120	4	2.46	11.2	1.71 × 10^9^

**Table 4 molecules-29-00363-t004:** Results of model fitting and lack of fit test.

Results	Model		
	Linear	2FI	Quadratic
Expansion ratio			
*R* ^2^	0.8428	0.8493	0.9867
*p*-value of lack of fit test	0.0040	0.0022	0.0814
Mean pore size			
*R* ^2^	0.7506	0.7583	0.9978
*p*-value of lack of fit test	<0.0001	<0.0001	0.1702

**Table 5 molecules-29-00363-t005:** Quadratic equation for response surfaces.

Response	Equation	AARD (%)
Expansion ratio (-)	22.42 + 0.145 A − 0.559 B − 0.316 C − 0.0003 A^2^ + 0.003007 B^2^ + 0.0082 C^2^ − 0.00064 AB + 0.00231 BC	7.1
Mean pore size (μm)	389.9 − 7.43 A + 0.365 B − 1.95 C + 0.03654 A^2^ + 0.003 B^2^ + 0.55 C^2^ − 0.00656 × 10^−3^ AB − 0.0225 BC	6.0

**Table 6 molecules-29-00363-t006:** Validation experiments for designing TPU 85A microcellular foam.

Exp. No.	*P* (bar)	*T* (°C)	*t* (h)	Expansion Ratio (-)		Pore Size (μm)	
				Exp. ^(a)^	Pred. ^(b)^	Exp. ^(a)^	Pred. ^(b)^
C1	120	140	4.7	5.54	5.69	12.7	12.4
C2	80	140	6	6.76	5.98	60.6	55.0

^(a)^ Experimental value. ^(b)^ Predicted value from equation listed in Table 5.

**Table 7 molecules-29-00363-t007:** The results of ANOVA for expansion ratio and mean pore size.

Source	*p*-Value ^(a)^	
	Expansion Ratio (-)	Mean Pore Size (μm)
Model	<0.0001	<0.0001
A	0.1044	<0.0001
B	<0.0001	<0.0001
C	0.6186	0.1393
A^2^	0.4277	<0.0001
B^2^	<0.0001	0.0663
C^2^	0.8245	0.0052
AB	0.1241	0.0023
AC	1.0000	0.9657
BC	0.5464	0.1532

^(a)^ *p*-value less than 0.05 indicates the term is significant at a 95% confidence level.

## Data Availability

The data presented in this study are available on request from the corresponding author.

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
