# Peer review of "The Application of Box–Behnken Design for Investigating the Supercritical CO2 Foaming Process: A Case Study of Thermoplastic Polyurethane 85A"

_molecules, 2024, doi:10.3390/molecules29020363_

Round 1

Reviewer 1 Report

Comments and Suggestions for Authors

Authors report the foaming of the Polyurethano 85A (PTU) with supercritical CO2, studying the effect of the different parameters with Box-Behnken design (BBD) to define a proper model to estimate pore radius and expansión rate.

As a general comment, the manuscript is well organized and the experimental procedure is well explained. However, the main drawback of this work is its novelty, since PTUs have been foamed many times with supercritical CO2 and BBD models have been also used many times in supercritical applications. In spite of this fact, this manuscript is interesting and can be published after carrying out more experiments to show if this model can be extrapolated to more PTU systems.

Comments:

There are more parameters that can be important in the foaming process to control pore size and expansion rate. Depressurization rate can modify pore size. Moreover, how the glass transition of the material is modified with the process must be addressed.

Authors perform an extrusion process to obtain the PTU precursor to be foamed. What is the geometry of the final piece before foaming?

The authors indicated that the inmersion time is not important and also they wrote that they did not observe important changes if the inmersion time was 6 hours. However, more experiments have to be done with the inmersion times and different conditions to justifiy this negligible effect.

The model results were successfully extrapolated with additional foaming PTU experiments. However, in order to confirm that this model can be used with other PTU systems, authors must checked this model with other published results concerning PTU foaming.

Finally, as a minor comment, in the introduction (lines 38-67) some references are cited. Nevertheless, the main results and novelties of these works are not included, and should be included in these paragraphs.

Reviewer 2 Report

Comments and Suggestions for Authors

The manuscript presents details of a Box-Behnken approach to investigate the performance of scCO2 foaming process for TPU 85A. Generally, the MS might be useful and of interest to the readers of MDPI Molecules, but several additional clarifications and modifications are needed before accepting the manuscript:

!. Aside from just reporting the results within the chosen investigation scheme, it would be important to add some more analyses of the used experimental conditions and data, so as to substantiate the specific choice of the (quadratic) model.

2. In order to reach wider audiences, it would be worthwhile to give more details on e.g. the choice of the temperatures and immersion times as presented in Tables 1 and 3, and about their link to structure-property relationships.

3. Although a previous paper about the procedure for physical-property analysis was mentioned (Ref.[22]), it would be advisable to state at least the key elements of this analysis.

4. Insofar as the results of ANOVA are linked to the so-called p-Value (Table 7), it would be interesting and valuable to comment the design of the proposed approach and the obtained results in view of recently published Editorial: Ronald L. Wasserstein & Nicole A. Lazar (2016) The ASA's Statement on p-Values: Context, Process, and Purpose, The American Statistician, 70:2, 129-133, DOI: 10.1080/00031305.2016.1154108

4. Some minor issues: Please explain “decreased melt strengths“ (line 186), “stable expansion ratio was finally reported” (line 273).

Comments on the Quality of English Language

Minor editing of English language required.

Round 2

Reviewer 1 Report

Comments and Suggestions for Authors

Authors have answered appropriately answered reviewer´s comments and the manuscript has been improved. The manuscript can be now accepted in its present form.

Reviewer 2 Report

Comments and Suggestions for Authors

Accept in present form.